# Nucleus, Cytoskeleton, and Mitogen-Activated Protein Kinase p38 Dynamics during In Vitro Maturation of Porcine Oocytes

**DOI:** 10.3390/ani9040163

**Published:** 2019-04-13

**Authors:** Payungsuk Intawicha, Li-Kuang Tsai, Shih-Ying Yen, Neng-Wen Lo, Jyh-Cherng Ju

**Affiliations:** 1Department of Animal Science, National Chung Hsing University, Taichung 40227, Taiwan; payungsuk@hotmail.com (P.I.); shiingyen@gmail.com (S.-Y.Y.); 2Division of Animal Science, School of Agriculture and Natural Resources, University of Phayao, Phayao 56000, Thailand; 3Bachelor Program of Biotechnology, National Chung Hsing University, No. 250, Kuokuang Rd., Taichung 402, Taiwan; leotsai123@hotmail.com; 4Department of Animal Science and Biotechnology, Tunghai University. 181, Sec. 3, Taichung Harbor Road, Taichung 407, Taiwan; nlo@thu.edu.tw; 5Graduate Institute of Biomedical Sciences, China Medical University, Taichung 40402, Taiwan; 6Translational Medicine Research Center, China Medical University Hospital, Taichung 40402, Taiwan; 7Department of Bioinformatics and Medical Engineering, Asia University, Taichung 41354, Taiwan

**Keywords:** cytoskeleton, IVM, MAPK, p38, oocyte, porcine

## Abstract

**Simple Summary:**

The exact roles played by p38 during oocyte maturation are not completely known and is the main theme of this study. The subcellular localization of p-p38 was increased along the progression of porcine oocyte maturation. Alterations in p38 expression and activation appeared to participate in regulating oocyte maturation, along with the progressive reorganization of the cytoskeleton and redistribution of cytoplasmic p-p38.

**Abstract:**

The mitogen-activated kinase (MAPK) p38, a member of the MAPK subfamily, is conserved in all mammalian cells and plays important roles in response to various physiologic cues, including mitogens and heat shock. In the present study, MAPK p38 protein expression in porcine oocytes was analyzed during in vitro maturation (IVM) by Western blotting and immunocytochemistry. The levels of p-p38 or activated p38 and p38 expression were at the lowest in the germinal vesicle (GV) stage oocyte, gradually rising at the germinal vesicle breakdown (GVBD) and then reaching a plateau throughout the IVM culture (*p* < 0.05). Similarly, the expression level of total p38 was also lower in the GV oocyte than in the oocyte of other meiotic stages and uprising after GVBD and remained high until the metaphase III (MII) stage (*p* < 0.05). In the GV stage, phosphorylated p38 (p-p38) was initially detectable in the ooplasm and subsequently became clear around the nucleus and localized in the ooplasm at GVBD (18 h post-culture). During the metaphase I (MI) and metaphase II (MII) stages, p-p38 was evenly distributed throughout the ooplasm after IVM for 30 or 42 h. We found that the subcellular localization increased in p-p38 expression throughout oocyte maturation (*p* < 0.05) and that dynamic reorganization of the cytoskeleton, including microfilaments and microtubules, was progressively changed during the course of meiotic maturation which was likely to be associated with the activation or networking of p38 with other proteins in supporting oocyte development. In conclusion, the alteration of p38 activation is essential for the regulation of porcine oocyte maturation, accompanied by the progressive reorganization and redistribution of the cytoskeleton and MAPK p38, respectively, in the ooplasm.

## 1. Introduction

The mitogen-activated kinase (MAPK) p38 is a member of the MAPK subfamily involved in a variety of cellular responses to cellular cues ranging from regulation of the cell cycle, cell death, and differentiation [1,2,3]. It has been reported that p38-MAPK, including the four isoforms (p38α, p38β, p38γ, and p38δ), plays essential roles in myogenic differentiation [4], breast cancer, and cell survival or apoptosis depending on the cell types and their environmental conditions [5,6]. 

In the ovary, oocytes are arrested at the prophase of first meiotic division, termed the germinal vesicle (GV) stage. Maturation of vertebrate oocytes involves a wide variety of signalings within oocytes, largely via activation and inactivation of specific protein kinases by phosphorylation and/or dephosphorylation procedures. In mammals, the GV starts to break down (GVBD) with gradual condensation of chromosomes progressing toward maturation in response to the luteinizing hormone (LH) surge [7]. This process is mainly regulated by maturation-promoting factor (MPF), a heterodimer of cyclin B (a regulatory subunit) and p34cdc2 (a catalytic kinase subunit) [8] and MAPK. Together with its downstream substrates, MAPK could be one of the cytostatic factors believed to stabilize MPF and to facilitate the MPF-driven progression of meiosis [9]. When the chromosomes are fully condensed, the oocyte enters metaphase I (MI) stage with a well-organized spindle, followed by anaphase I and telophase I, and is then arrested in the metaphase II (MII) stage, at which the activation by sperm or other artificial stimuli could normally occur [8]. The reorganization or redistribution of the endoplasmic reticulum and likely some other organelles, such as mitochondria, is required to prepare oocytes for Ca^2+^ oscillations that trigger oocyte activation during fertilization [10]. 

Other members of the MAPK family that are involved in meiotic maturation of oocytes are the extracellular regulated kinases (ERKs) [11], such as ERK1 and ERK2, which are members of the canonical MAPK family activated in response to various extracellular signaling molecules via an upstream small G-protein Ras [12]. In addition, other MAPK members Jun kinases (JNK) and p38, collectively known as stress-activated protein kinases, can also be induced by extracellular cues [13,14]. These signaling pathways are also known to play critical roles in regulation and determination of cell growth, proliferation, differentiation, and apoptosis under physiologic and stress conditions, during which morphological and subcellular alterations may be observed in animal cells to maintain vital physiological functions [15]. In mammalian oocytes, similar changes associated with the reorganization of cytoskeleton, including microtubules and microfilaments, driven by small GTPase RhoA during the process of meiotic maturation or subsequent cell division can occur [16,17]. 

In porcine species, p38 MAPK also exerts a broad spectrum of physiologic roles, including the control of actin polymerization in smooth muscle cells that is necessary for platelet-derived growth factor-induced formation of lamellipodia, a dynamic cell structure composed of three zones of highly organized filamentous actin, for cell migration [18,19]. Although p38 appears to participate in oocyte maturation [20] and preimplantation development of murine embryos [21], its significance and definitive roles in porcine oocyte maturation are largely unknown. Therefore, the present study aimed to investigate the dynamics of p38 level and its localization in the ooplasm as the first step to understand its roles in oocyte maturation. Dynamic reorganization of the cytoskeleton was also examined as the confirmation of maturation stages along the meiotic progression of porcine oocytes.

## 2. Materials and Methods

### 2.1. Chemicals

All chemicals and antibodies used were purchased from Sigma-Aldrich (St. Louis, MO, USA), Santa Cruz Biotechnology Inc. (Santa Cruz, CA, USA), or Cell Signaling Technology Inc. (Danvers, MA, USA) unless mentioned otherwise.

### 2.2. Oocyte Collection and In Vitro Maturation (IVM)

Porcine ovaries were collected from prepubertal gilts, stored in saline (35–37 °C), and transported to the laboratory in an insulated thermal container within 1 h after slaughter. Cumulus–oocyte complexes (COCs) were harvested by aspirating ovarian surface follicles (3 to 6 mm in diameter), and matured in North Carolina State University-23 (NCSU-23) medium supplemented with 10% follicular fluid, cysteine (0.1 mg/mL), epidermal growth factor (EGF; 10 ng/mL) and gonadotropins (human chorionic gonadotropin (hCG), 10 IU/mL; pregnant mare’s serum gonadotropin (PMSG), 10 IU/mL). In each 50 μL microdrop of IVM medium overlaid with mineral oil, 20–30 COCs were cultured for the first 22 h of culture at 39 °C in an incubator containing 5% CO_2_ in air [22]. Thereafter, the medium was replaced with gonadotropin-free NCSU-23 medium, and the oocytes were then cultured continuously for another 20 h. Matured oocytes were selected by visualization of polar body extrusion at 42 h after the onset of IVM [23].

### 2.3. Immunocytochemistry

#### 2.3.1. Nuclear and Cytoskeletal Progression

Oocytes were washed twice in DPBS-PVA (Dulbecco’s phosphate-buffered saline containing 1% (v/v) polyvinyl alcohol) and then fixed in DPBS-PVA containing 4% paraformaldehyde and 0.2% Triton X-100 for 40 min at room temperature. Thereafter, fixed samples were washed twice in DPBS-PVA for 15 min and stored overnight in 1% bovine serum albumin (BSA) in DPBS-PVA (BSA-DPBS-PVA) at 4 °C prior to staining. In the following day, oocytes were blocked with 10% goat serum (Dako A/S, Glostrup, Denmark) in BSA-DPBS-PVA for 45 min and then incubated in BSA-DPBS-PVA containing antitubulin primary antibodies at 4 °C overnight. Then, they were washed three times with washing solution containing 2% BSA, 2% goat serum, 0.2% milk powder, 0.2% sodium azide, and 0.1% Triton in DPBS. The oocytes were subsequently incubated with the fluorescein isothiocyanate (FITC)-conjugated secondary antibody at 39 °C for 2 h, then washed three times, as with primary antibodies. Microfilaments of the oocytes were stained with rhodamine phalloidin (Molecular Probe, R415) for 1 h to label the filamentous actin (F-actin) and then washed three times again. Finally, the oocytes were mounted on the slide with mounting medium containing Hoechst 33342 and then sealed with clear fingernail polish. 

#### 2.3.2. Subcellular Localization of Phosphorylated p38 (p-p38)

Fixed oocytes were incubated with rabbit polyclonal anti-phospho-p38 antibody (1:100, #9211) at 4 °C overnight. After three washes in BSA-DPBS-PVA, oocytes were incubated in BSA-DPBS-PVA containing Alexa Fluor 488-labeled goat anti-rabbit IgG (1:300; Molecular Probes Inc., Eugene, OR, USA) for 40 min at room temperature, and the chromosomes were then stained with Hoechst 33342 (10 μg/mL). Negative control images were obtained by omitting the first antibody during staining. Following complete washing, oocytes were mounted on slides with mounting medium (50% DPBS, 50% Glycerol, 25 mg/mL NaN3) and observed under an Olympus epifluorescence microscope (AX-70). The intensity of p-p38 expression in oocytes was analyzed with Image J software [24].

### 2.4. Western Blotting for p38 Analysis

Analysis of protein expression was conducted as described previously [7,25]. Oocytes (*n* = 150 were first rinsed in DPBS-PVA immediately after treatments and then collected in sodium dodecyl sulfate (SDS) sample buffer containing 100 mM Tris/HCl (pH = 6.8), 5% 2-mercaptoethanol, 3% SDS, 4% glycerol, and 0.1% bromophenol blue. All samples were boiled for 5 min and then stored at −80 °C before analysis. Oocyte samples were subjected to electrophoresis in 10% (v/v) polyacrylamide/SDS gels [7,24]. The resolved proteins were transferred to nitrocellulose membranes, blocked with 10% chicken serum in Tris-buffered saline (20 mM Tris-HCl, pH 7.4, 150 mM NaCl) containing 0.1% Tween-20 for 1 h, and then incubated with rabbit polyclonal anti-phospho-p38 (1:100, #9211, Cell signaling) and anti-p38 antibodies (1:100, #9212, Cell signaling), respectively, at 4 °C overnight. Membranes were washed three times (10 min/wash) with Tris-buffered saline Tween-20 (20 mM Tris, pH 7.4, 500 mM NaCl, 0.05% Tween-20) and then incubated with secondary antibody (1:10000, anti-rabbit immunoglobulin horseradish peroxidase) for 1 h at room temperature. After three washes (10 min each), proteins were detected by the Super^®^ Signal West Pico Chemiluminrescent Substrate Kit (Pierce Biotechnology, Inc., Rockford, IL, USA). Band intensities, including p38 and phosphorylated-p38, were measured using Image J software and normalized to total-p38 before statistical analysis.

### 2.5. Experimental Designs

#### 2.5.1. Experiment 1: Nuclear and Cytoskeletal Progression of Maturing Porcine Oocytes 

Oocytes at GV (0 h), GVBD (18 h), MI (30 h), and MII (42 h) stages were stained for cytoskeleton, i.e., microfilaments and microtubules, and then were examined under an epifluorescence microscope.

#### 2.5.2. Experiment 2: Subcellular Localization of p-p38 in Porcine Oocytes during IVM

Oocytes were studied at the GV (0 h), GVBD (18 h), MI (30 h), and MII (42 h) stages. After fixation, oocytes were immunostained for p-p38 and then observed under an epifluorescence microscope.

#### 2.5.3. Experiment 3: Expression of p38 and p-p38 in Porcine Oocytes

Oocytes at the GV (0 h), GVBD (18 h), MI (30 h), and MII (42 h) stages were subjected to electrophoresis in 10% (v/v) polyacrylamide/SDS gels. The resolved proteins were transferred to nitrocellulose membranes and incubated with antibody for detection of band intensities.

### 2.6. Statistical Analyses

All data from Western blotting and immunocytochemical staining were analyzed by ANOVA using the general linear model (GLM) procedure in the Statistical Analysis System [26], followed by Tukey’s test. Percentile data were analyzed by completely randomized designs. For all statistical analyses, significance level was set at *p* < 0.05.

## 3. Results

### 3.1. Nuclear and Cytoskeletal Progression of Maturing Porcine Oocytes

Morphological classification based on the chromatin structure and cytoskeletal characteristics of GV stage porcine oocytes was described. We found three distinct types of porcine oocytes at the GV stage. In Type I GV oocytes, the most typical configuration of the GV was its chromatin being clearly visible and dispersed in the GV nucleus (Figure 1A). The interphase microtubular network (Figure 1B) and microfilaments (Figure 1C) were uniformly spread over the periooplasmic area. The Type II GV oocytes possessed moderately condensed and less dispersed nuclear material. Some perinucleolar chromatin structures could be observed. (Figure 1D). Structures of microtubules and microfilaments were similar to those seen in the Type I oocytes (Figure 1E,F). In the Type III GV oocytes, the chromatin displayed single nucleolus-like configuration with only little chromatin material in the GV nucleus (Figure 1G). The periooplasmic microtubules were similar to those in the Type I GV oocytes (Figure 1H) with no discernible changes in the microfilament pattern (Figure 1I).

When GVBD occurred, the chromatin gradually condensed into chromosomal structures (Figure 2A). Cytoplasmic microtubules formed a spindle-like structure in the ooplasm with little or no periooplasmic microtubules discernible (Figure 2B); reduced microfilament intensity (Figure 2C) was also observed. In the MI stage, chromosomes were aligned on the equatorial plate (Figure 2D). The spindle structure was formed by microtubules in the ooplasm, while microfilaments formed distinct actin caps on the periooplasmic area immediately adjacent to the spindle (Figure 2F). In the MII stage, the first polar body was extruded (arrow) and chromosomes aligned on the equatorial plate (Figure 2G). The spindle structure was formed with the decreasing periooplasmic microtubules, and the intensity of the microfilaments increased between the polar body and the mature oocyte (Figure 2I).

### 3.2. Subcellular Localization of p-p38 in Porcine Oocytes during IVM

Expression of p-p38 in GV oocytes was lower than that in other stages; it gradually increased after GVBD and remained in the plateau to the MII stage (Figure 3A). Activation and subcellular localization of p-p38 in porcine oocytes during meiotic maturation were quantified (Figure 3B, *p* < 0.05). At the GV stage, p-p38 was initially detectable in the ooplasm (Figure 4A) and subsequently surrounded the nucleus (Figure 4B) around GVBD (18 h post-culture). In the MI and MII stages, p-p38 was evenly distributed throughout the ooplasm (Figure 4C,D) after culture for 30 or 42 h. 

### 3.3. Expressions of p38 and p-p38 in Porcine Oocytes during IVM

With the immunoblot, we found that p38 and p-p38 expressions were increased during IVM culture (*p* < 0.05). The expression of total p38 was the lowest (*p* < 0.05) in the GV stage oocytes. Relative activity of p38 increased at the GVBD (*p* < 0.05) and remained high at MI and MII stages. A representative immunoblot of p38 and p-p38 expressions is shown in Figure 5A. The activated p38 level (p-p38) was the lowest in GV oocytes; it started to increase at GVBD and then sustained at a high level throughout the course of IVM (Figure 5B, *p* < 0.05).

## 4. Discussion

For mammalian ovigenesis, it is well known that oocytes from small follicles possess smaller oocytes (107.2 ± 0.6 μm in diameter) than those obtained from large follicles, where oocytes have a relatively large diameter (116.9 ± 0.7 μm). In vitro culture of these small oocytes generally has a lower maturation rate when compared to large follicles [14]. This known fact is largely attributed to the immaturity of the nucleus and the ooplasm of the growing oocytes. In contrast, the dynamic changes of oocyte nucleus and cytoskeleton, including microfilaments and microtubules, in association with oocyte maturation stages have not been thoroughly investigated. For those fully grown oocytes, accompanied by the reorganization or redistribution of organelles, the cytoskeleton and the nucleus of maturing oocytes undergo significant changes or alterations during IVM. In the present study, three nuclear types of GV stage oocytes were classified. In Type I GV oocytes, the most typical configuration of the GV was its clearly visible chromatin dispersed in the GV (Figure 1A). Type II GV oocytes were characterized by moderately condensed and less dispersed nuclear material in the nucleus. In the Type III GV oocytes, the chromatin displayed single nucleolus-like configuration with only very little chromatin material visible in the GV nucleus (Figure 1G). Our results are similar to previous studies [27,28].

It is known that p38 MAPK signaling is crucial for preimplantation development [29] and is also associated with cellular apoptosis [30]; without it, embryos display reversible development arresting at the 8–16 cell stages or beyond [29]. Nevertheless, little information regarding p38 signaling during oocyte maturation has been reported in pig oocytes [7,31]. In the present study, we first found that p-p38 level could be detected in the GV stage oocytes by immunoblotting; therefore, a linear increase in the expression of p-p38 levels in the ooplasm with progression of IVM was observed (Figure 3). Furthermore, when the total expression of p38 was also considered, the ratios of p-p38 to total p38 level (fold changes) for each maturation stage of oocytes were calculated. A slightly different patterns of increase was revealed, where the relative p-p38 expressions after GVBD were all significantly increased with no detectable differences found among different maturation stages (Figure 5), indicating that the total level of p38 expression in porcine oocytes also increased proportionally during IVM, particularly by the end of the maturation stage. We therefore reasoned that, in addition to MPF, MAPK p38 may also actively participate in the maturation process of porcine oocytes; however, this assumption requires more investigation. 

Similarly, p38 kinase initially accumulated in the GV stage oocyte and then became activated (by phosphorylation) and dispersed concurrently in and/or around the nucleus by GVBD. It is apparent that the p-p38 translocated from the perinuclear region to the ooplasm while reaching maturity (Figure 4). It has also been reported that p38 protein mainly resides in the nucleus before GVBD and is then progressively localized in the ooplasm and close-by chromosomes at MI to MII stages of porcine oocytes [32], which supports our observation in this study. Moreover, matured oocytes with a high level of p-p38 were also observed in that previous study; therefore, it is most likely that the activation of p38 is involved in porcine oocyte maturation and subsequent embryo development under nonstress conditions [7]. However, it has been known that activation of the p38 pathway can also promote phosphorylation of a small heat shock protein 27 [33,34], which is catalyzed by MAPK-activated protein kinase-2 (MAPKAPK-2), a serine–protein kinase immediately downstream of p38 under stressful culture conditions [35,36]. In turn, phosphorylation of heat shock protein 27 can lead to stabilization of cellular actin filaments against environmental insults to modulate the dynamic changes of microfilaments in response to p38 activation [37,38]. However, when cellular heat shock protein 27 concentration is abnormally elevated, it can cause erroneous actin polymerization, leading to cell blebbing or apoptosis, as reported in previous studies [39,40].

p38 MAPK has been reported to be closely related to cellular apoptosis [15,41,42]; however, it has also been found to be associated with oocyte maturation similar to MPF in pig and mouse oocytes during maturation, as reported in the present and previous studies [43,44]. Presumably, in vitro culture systems are also, to a certain degree, stressful or suboptimal to the culture cells, including oocytes and embryos [7,21,25]. Therefore, p38 expression and/or activation (phosphorylation) disclosed in the current study are also likely to play dual or multiple roles, for instance, oocyte maturation, apoptosis, and/or cytoskeleton reorganization. However, more delicate experimentation may be required to reveal how an oocyte maintains such a delicate homeostasis between these cellular events.

## 5. Conclusions

The present study found that spatial and temporal alterations of p38 activation are necessary for the regulation of porcine oocyte maturation. Nuclear and cytoskeletal reorganizations are well defined, and they are closely related to oocyte maturation and activation of p38 MAPK in porcine species. Nevertheless, how p38 elaborates its homeostatic switch between oocyte maturation and apoptosis, as well as the interactions with MPF and cytoskeleton, if any, deserves further investigation. 

## Figures and Tables

**Figure 1 animals-09-00163-f001:**
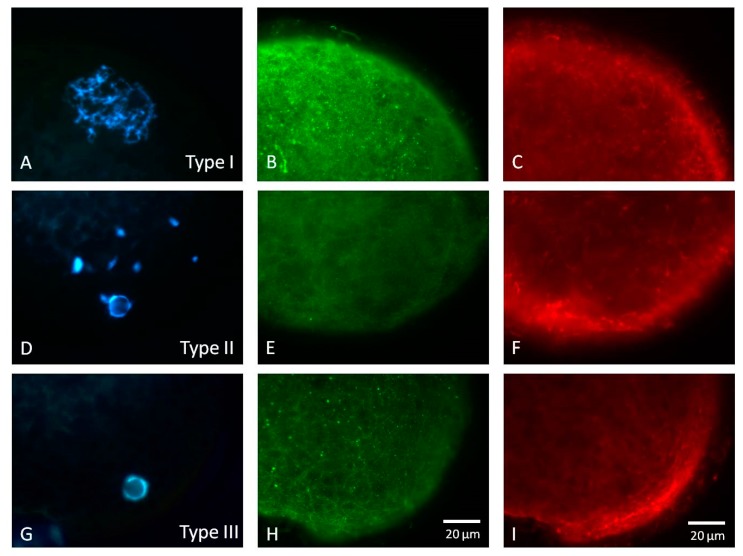
Morphological classification of the chromatin and the cytoskeleton of porcine oocytes at the germinal vesicle stage (GV). In Type I GV oocytes, the most typical configuration of the GV was its chromatin being clearly visible and dispersed in the GV nucleus (**A**). The interphase microtubular network (**B**) and microfilaments (**C**) uniformly spread over the periooplasmic area. The Type II GV oocytes possessed moderately condensed and less dispersed nuclear material. Some perinuclear chromatin structure could be observed (**D**). Ultrastructures of microtubules and microfilaments were similar to those seen in the Type I oocytes (**E**,**F**). In the Type III GV oocytes, their chromatin displayed single nucleolus-like configuration with only little chromatin material in the GV nucleus (**G**). The periooplasmic microtubules were similar to those in Type I GV (**H**) with no discernible changes in the microfilament pattern (**I**). Scale bar, 20 μm.

**Figure 2 animals-09-00163-f002:**
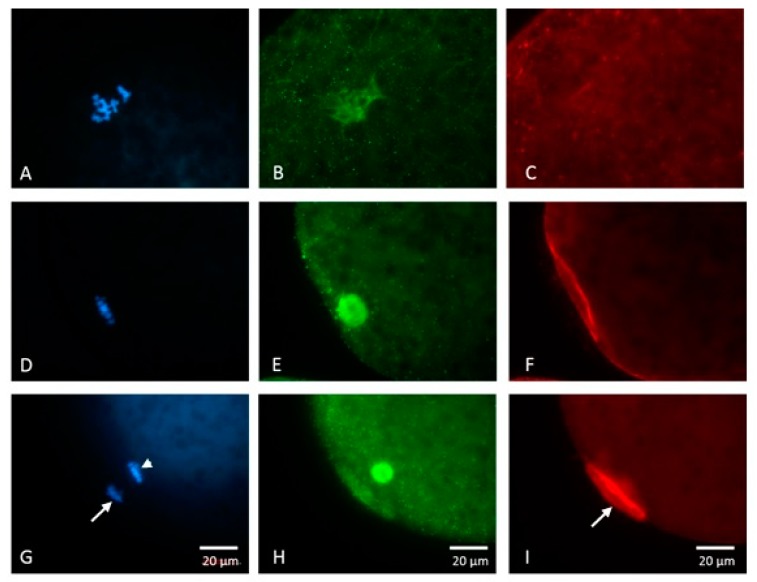
Nuclear and cytoskeletal dynamics of porcine oocytes during in vitro maturation (IVM). Germinal vesicle breakdown (GVBD) and chromatin condensed into chromosomes around 18 h after the onset of IVM (**A**). Microtubules transformed into a spindle-like structure in the ooplasm with little or no periplasmic microtubules (**B**) during which reduced microfilament intensity (**C**) was visible. In metaphase I (MI), chromosomes aligned on the equatorial plate (**D**). Meiotic spindle structure was formed by microtubules in the ooplasm, while microfilaments became thickened on the periooplasmic area near the MI spindle (**F**). (**G**) In the metaphase II (MII) stage, a polar body was extruded (arrow) and chromosomes realigned on the equatorial plate (arrowhead) with the formation of the MII spindle. (**H**) The intensity of microfilaments increased between the polar body and the matured oocyte, forming an actin cap-like structure (**I**, arrow). Scale bar, 20 μm.

**Figure 3 animals-09-00163-f003:**
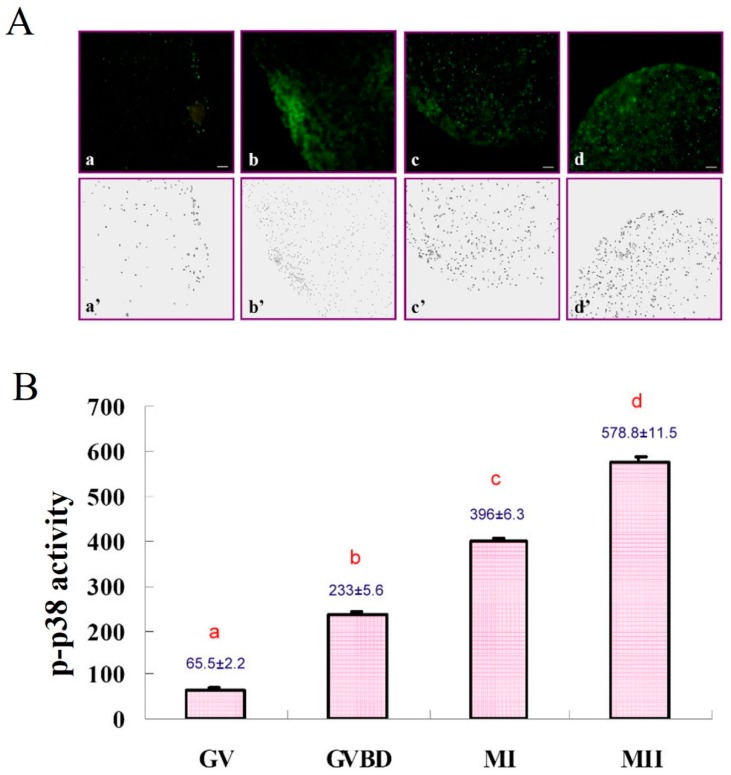
Fluorescence intensity of p-p38 activity in porcine oocytes during IVM. (**A**) a–d, expressions of p-p38 in the oocytes after the onset of IVM for 0 h (a, GV), 18 h (b, GVBD), 30 h (c, MI), and 42 h (d, MII). a’–d’, the expression of p-p38 quantified by Image J software (National Institutes of Health: Bethesda, MA, USA, 2006). Scale bar. (**B**) Fluorescence intensity of p-p38 significantly increased with progression of oocyte maturation (*p* < 0.05). a–d, bars without a common superscript differed (*p* < 0.05). Bars are means ± SEM (five replicates, with five oocytes/replicate).

**Figure 4 animals-09-00163-f004:**
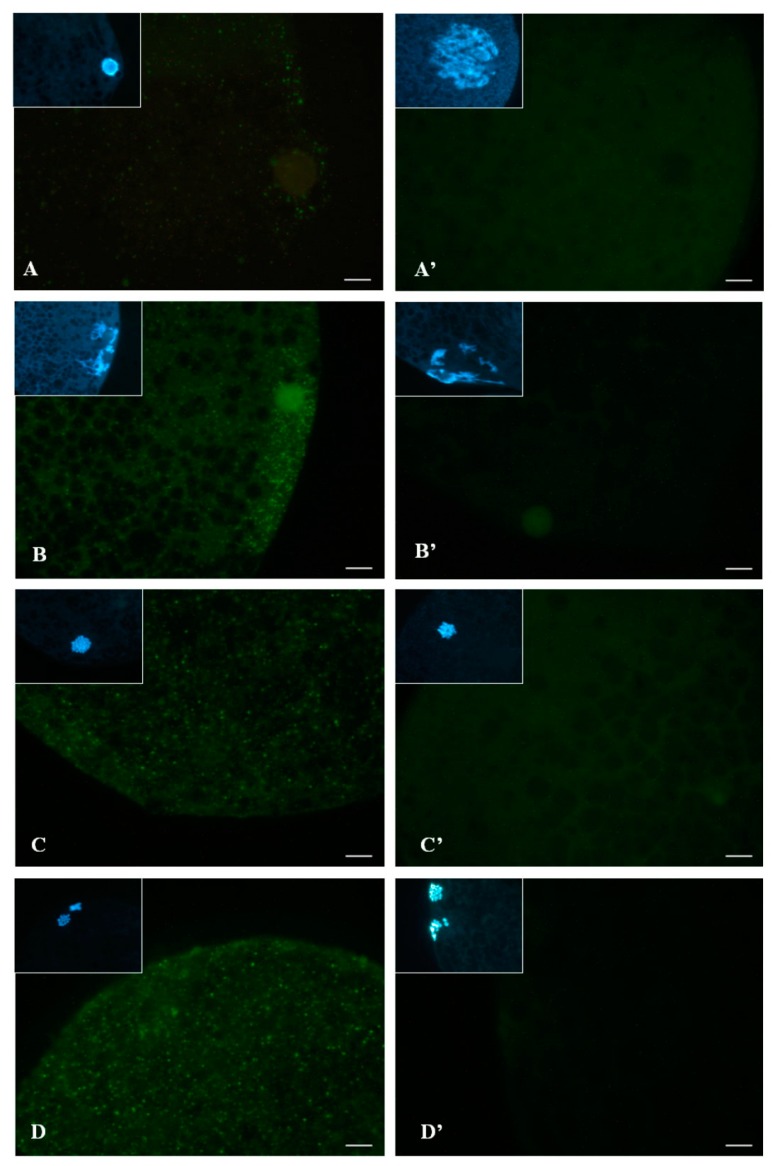
Subcellular localization of phosphorylated p38 (p-p38) in porcine oocytes during IVM. (**A**) Phosphorylated p38 was initially visible as a few green dots in the ooplasm at the GV stage prior to IVM culture (0 h). (**B**) p-p38 was somewhat more concentrated in or around the GV nucleus at GVBD 18 h after IVM. At the MI (**C**) and MII (**D**) stages, p-p38 was localized to the ooplasm surrounding chromosomes. A’ to D’ are the negative controls. Insets show chromatin or chromosomes (blue) stained with Hoechst 33342. Scale bar, 10 μm.

**Figure 5 animals-09-00163-f005:**
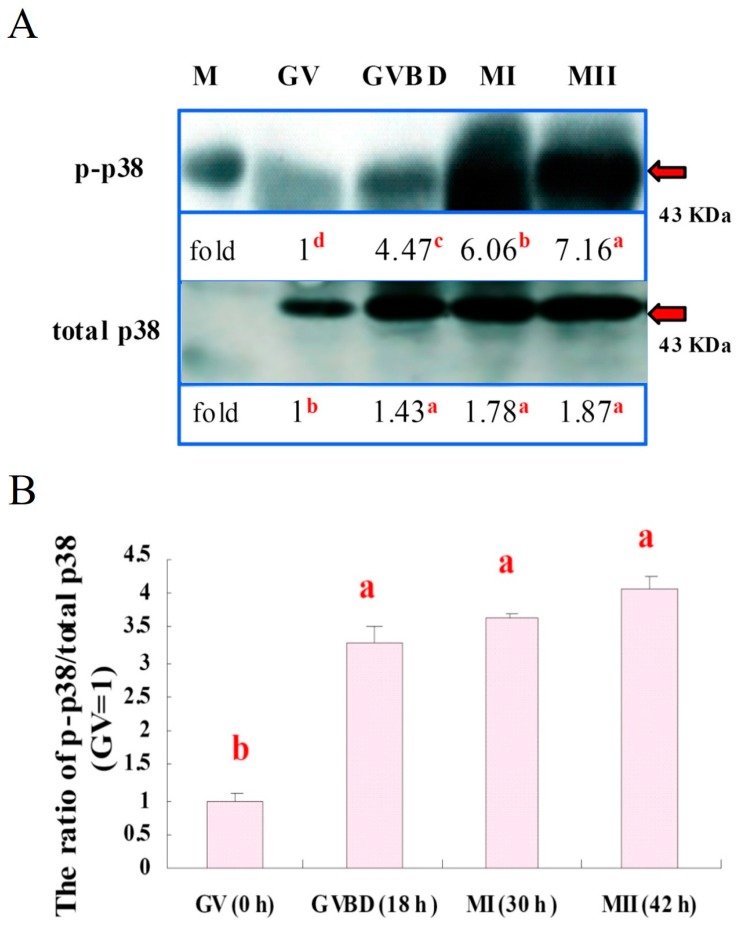
Expressions of p38 and phosphorylated p38 (p-p38) in porcine oocytes during IVM. (**A**) A representative immunoblot of total p38 and p-p38; the expression of p-p38 increased progressively during the IVM (*p* < 0.05). The expressions of both total p38 and p-p38 were the lowest (*p* < 0.05) in the GV stage. (**B**) Relative activity of p38, however, increased at the GVBD (*p* < 0.05) and remained high at the MI and MII stages. Each lane of the SDS-PAGE gel contained 150 oocytes (three replicates). a–d, lanes or bars without a common superscript differed (*p* < 0.05). Bars are means ± SEM. M: marker.

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
