# Peer review of "Nucleus, Cytoskeleton, and Mitogen-Activated Protein Kinase p38 Dynamics during In Vitro Maturation of Porcine Oocytes"

_animals, 2019, doi:10.3390/ani9040163_

Round 1

Reviewer 1 Report

Major comments to animals-454779-original V0

Through the manuscript, there are many careless mistakes, therefore the authors have to more carefully prepare the manuscript for publication. The greater part of Discussion does not appear to be appropriate.

Specific comments

1. Line 22: “of” needs to be inserted between “reorganization” and “the”.

    2. Authors used P32” and “p32”. Is there a reason?

3. Lines 47-49: A sentence, “It has been reported….”, is complicated and confusing. It needs to be rewritten.

4. Lines 57-58: “a regulatory subunit” should be “(a regulatory subunit)”.

5. Line 58: “mitogen-activated protein kinase” was already abbreviated in line 45.

6. Line 60: “cytostatic factors” was never used thereafter, therefore it needs not to be abbreviated.

7. Line 64: See comment #7 for “endoplasmic reticulum (ER)”.

8. Line 71: See comment #7 for “(SAPKs)”.

9. Line 80: See comment #7 for “(MAPs)”.

10. Line 84: See comment #7 for “(PDGF)”.

11. Lines 101-105: A sentence, “The recovered COCs….”, is confusing and should be rewritten.

12. Lines 111-127: This paragraph should be carefully rewritten. For example, what is PVA?  Once 1% BSA in Dulbecco's phosphate-buffered saline-PVA” was abbreviated as BSA-DPBS-PVA, that should be used consistently. This reviewer guesses “Dulbecco's phosphate-buffered saline” should be consistently described as “DPBS”. What is happened in lines 118-119? Authors should more carefully write the manuscript for publication.

14. Line 138: “2.3” should be “2.4”.

15. Line 153: “Subs” should be “Substrate”.

16. Section 2.5: Many sentences can be seen in section 2.2-2.4, thus they are repetitive. Section 2.2-2.5 can be reorganized and rewritten.

17. Figure 3: (a) and (b) should be A and B, respectively. What is “M” in Fig. 3B?

18. Figure 5: (a) and (b) should be A and B, respectively.

19. Discussion: The greater part of “Discussion”, e.g. 2nd and 3rd paragraph, cannot be tightly connected with results obtained in this report, which is confusing this reviewer very much. Discussion is alien to the results and conclusions.

Author Response

Reviewer 1

Open Review

English language and style

(x) Extensive editing of English language and style required 
( ) Moderate English changes required 
( ) English language and style are fine/minor spell check required 
( ) I don't feel qualified to judge about the English language and style 

Yes

Can be improved

Must be improved

Not applicable

Does the introduction provide sufficient background and include   all relevant references?

( )

(x)

( )

( )

Is the research design appropriate?

( )

(x)

( )

( )

Are the methods adequately described?

( )

(x)

( )

( )

Are the results clearly presented?

(x)

( )

( )

( )

Are the conclusions supported by the results?

( )

( )

(x)

( )

Comments and Suggestions for Authors

Major comments to animals-454779-original V0

Through the manuscript, there are many careless mistakes, therefore the authors have to more carefully prepare the manuscript for publication. The greater part of Discussion does not appear to be appropriate.

Specific comments

1.     Line 22: “of” needs to be inserted between “reorganization” and “the”.

Response: Thank you for the comment.  we have inserted of as shown on line 23.

2.     Authors used “P38” and “p38”. Is there a reason?

Response: Thank you for pointing it out.  We have changed all those into p38 throughout the MS.

3. Lines 47-49: A sentence, “It has been reported….”, is complicated and confusing. It needs to be rewritten.

Response: Thank you for the comments.  In the revised manuscript, we have changed it into “It has been reported that four isoforms of p38-MAPK (p38, p38, p38, and p38) are involved in myogenic differentiation [4], breast cancer, and cell survival [5]” as shown on line 49-51.

4. Lines 57-58: “a regulatory subunit” should be “(a regulatory subunit)”.

Response: Thank you for the comment.  we have revised as suggested (Line 59).

5. Line 58: “mitogen-activated protein kinase” was already abbreviated in line 45.

Response: It has been corrected as pointed out (Line 58-59).

6. Line 60: “cytostatic factors” was never used thereafter, therefore it needs not to be abbreviated.

Response: The abbreviation has been deleted as suggested (Line 60-61).

7. Line 64: See comment #7 for “endoplasmic reticulum (ER)”.

Response: “(ER)” has been deleted as suggested (Line 65). 

8. Line 71: See comment #7 for “(SAPKs)”.

Response: “(SAPKs)” has been deleted as suggested (Line 72). 

9. Line 80: See comment #7 for “(MAPs)”.

Response: “(MAP)” has been deleted as suggested (Line 81). 

10. Line 84: See comment #7 for “(PDGF)”.

Response: (PGDF)” has been deleted as suggested (Line 85). 

11. Lines 101-105: A sentence, “The recovered COCs….”, is confusing and should be rewritten.

Response: Thank you for the comments.  The sentences have rephrased as shown on line 103-107.

12. Lines 111-127: This paragraph should be carefully rewritten. For example, what is PVA?  Once “1% BSA in Dulbecco's phosphate-buffered saline-PVA” was abbreviated as BSA-DPBS-PVA, that should be used consistently. This reviewer guesses “Dulbecco's phosphate-buffered saline” should be consistently described as “DPBS”. What is happened in lines 118-119? Authors should more carefully write the manuscript for publication.

Response: The abbreviations have been defined and used thereafter (Line113-123 and Line131-137).

14. Line 138: “2.3” should be “2.4”.

Response: It has been corrected to “2.4”, and sorry for the mistake (Line 140).

15. Line 153: “Subs” should be “Substrate”.

Response: It has been revised as suggested (Line 154).

16. Section 2.5: Many sentences can be seen in section 2.2-2.4, thus they are repetitive. Section 2.2-2.5 can be reorganized and rewritten.

Response: Thank you for the comments.  These sections have been improved (Line158-176).

17. Figure 3: (a) and (b) should be A and B, respectively. What is “M” in Fig. 3B?

Response: Thank you for the help.  The mistakes have been corrected, and M stands for marker as added to the figure legend (We have switched Figure 3 and Figure 5).

18. Figure 5: (a) and (b) should be A and B, respectively.

Response: The mistakes have been corrected as shown in Figure 3 A and B. (We have change Figure 5 to Figure 3)

19. Discussion: The greater part of “Discussion”, e.g. 2nd and 3rd paragraph, cannot be tightly connected with results obtained in this report, which is confusing this reviewer very much. Discussion is alien to the results and conclusions.

      Response: Thanks for the comments. We have completely reorganized the order of Figure presentation (original Figures 3 and 5 are switched) and the context of Discussion as suggested (Line 291-349).

Reviewer 2 Report

The study describes the expression of p-38 and phosphorylated p-38 in the porcine oocyte during in vitro maturation.  An increase in the level of p38 and p-p38 in the oocyte was observed throughout maturation from germinal vesicle (GV) to metaphase II (MII). Relative activity of p-38, as assessed through the ratio of activated (phosphorylated p-38) to total p-38, was lowest in GV stage oocytes and increased from GVBD.  Changes in the subcellular localization of p-p38 within the oocyte are also described.   The results are of some interest, and add to the understanding of the potential role of p-38 activation in oocyte maturation.  

General comments:

While these findings support a role for p-38 activation in regulating porcine oocyte maturation, the conclusion that they prove that it is essential  (Line 39 and 319) seems a strong statement, given the nature of the study. 

It was not clear how novel the information on changes in nuclear chromatin configuration of the porcine oocyte during meiosis is.  Has anyone else reported the nuclear confirmation of GV stage porcine oocytes?  There appears to be minimal reference to other studies.

Methods: The authors have used methods they have experience with and have previously published.  The methods would benefit from a clear description of numbers of oocytes and replicates analysed.  These are reported in Figure legends, but should be incorporated into the methods and results.  The methods indicate that 200 oocytes were used for Western blotting, while the Figure legend reports 150 oocytes.

Results:  Line 211-213 appears to be discussing the immunoblot results, but is included in the section on subcellular localization. Should this sentence be in 3.2 rather than 3.3.

Line 213-214: suggests that expression levels of total p-38 are shown in Figure 5b, but this figure reports p-p38, not p-38 expression?

Line 215-216: suggests that Figure 5B reports the ratio of p-p38 / p-38, but this Figure reports p-38 expression, while Figure 3b reports the ratio.  This sentence also appears to relate to results from the immunoblot, and should be included in 3.2 rather than 3.3.

Discussion: The Discussion could be significantly improved.

The relevance of the final 2 sentences of the first paragraph is unclear (lines 269-272).  Oocytes from different size follicles were not studied.

Line 277 appears to suggest that this study was the first to detect p-p38 in porcine oocytes by immunoblotting; however, the authors have previously used this method to detect expression in MII oocytes (Yen et al, 2014).

The third paragraph of the Discussion essentially replicates discussion from the authors previous paper (Yen et al 2014) on the expression and activation of MAPK in porcine oocytes under heat stress.  While some points have relevance to the current study, as culture could be another form of stress, it is unclear whether replication of the whole paragraph discussing heat shock proteins  is warranted and appropriate.

The discussion appears to end prematurely - Line 317 is an incomplete sentence.

Other comments:

Line 20 and Line 34: It is not clear how subcellular localization of p-p38 can increase. Was the purpose of this sentence to comment on changes in the localization of p-p38, or to comment on the increase in p-p38 expression throughout oocyte maturation.

Line 21: "Alteration of p38 activation" is unclear, please reword.

Line 27: The level of activated p-38, please reword.

Line 27: immunocytochemistry

Line 118: "containing anti- and" please provide the full name of this antibody.

Line 129: Fixed oocytes, please correct spelling

Line 137: expression in oocytes

Line 137: Please provide more information on the analysis used for determining the intensity of expression.  Was the whole oocyte assessed, were controls in place to ensure staining intensity was similar between replicates?

Line 202: what is meant by "after prolonged IVM culture", the period described appears to be the standard period for IVM of porcine oocytes.

Figure 3a, why is no marker visible for the p38 blot.

Line 297: "during the prolonged heat shock" please remove the word protein.

Author Response

Reviewer 1

Open Review

English language and style

(x) Extensive editing of English language and style required 
( ) Moderate English changes required 
( ) English language and style are fine/minor spell check required 
( ) I don't feel qualified to judge about the English language and style 

Yes

Can be improved

Must be improved

Not applicable

Does the introduction provide sufficient background and include   all relevant references?

( )

(x)

( )

( )

Is the research design appropriate?

( )

(x)

( )

( )

Are the methods adequately described?

( )

(x)

( )

( )

Are the results clearly presented?

(x)

( )

( )

( )

Are the conclusions supported by the results?

( )

( )

(x)

( )

Comments and Suggestions for Authors

Major comments to animals-454779-original V0

Through the manuscript, there are many careless mistakes, therefore the authors have to more carefully prepare the manuscript for publication. The greater part of Discussion does not appear to be appropriate.

Specific comments

1.     Line 22: “of” needs to be inserted between “reorganization” and “the”.

Response: Thank you for the comment.  we have inserted of as shown on line 23.

2.     Authors used “P38” and “p38”. Is there a reason?

Response: Thank you for pointing it out.  We have changed all those into p38 throughout the MS.

3. Lines 47-49: A sentence, “It has been reported….”, is complicated and confusing. It needs to be rewritten.

Response: Thank you for the comments.  In the revised manuscript, we have changed it into “It has been reported that four isoforms of p38-MAPK (p38, p38, p38, and p38) are involved in myogenic differentiation [4], breast cancer, and cell survival [5]” as shown on line 49-51.

4. Lines 57-58: “a regulatory subunit” should be “(a regulatory subunit)”.

Response: Thank you for the comment.  we have revised as suggested (Line 59).

5. Line 58: “mitogen-activated protein kinase” was already abbreviated in line 45.

Response: It has been corrected as pointed out (Line 58-59).

6. Line 60: “cytostatic factors” was never used thereafter, therefore it needs not to be abbreviated.

Response: The abbreviation has been deleted as suggested (Line 60-61).

7. Line 64: See comment #7 for “endoplasmic reticulum (ER)”.

Response: “(ER)” has been deleted as suggested (Line 65). 

8. Line 71: See comment #7 for “(SAPKs)”.

Response: “(SAPKs)” has been deleted as suggested (Line 72). 

9. Line 80: See comment #7 for “(MAPs)”.

Response: “(MAP)” has been deleted as suggested (Line 81). 

10. Line 84: See comment #7 for “(PDGF)”.

Response: (PGDF)” has been deleted as suggested (Line 85). 

11. Lines 101-105: A sentence, “The recovered COCs….”, is confusing and should be rewritten.

Response: Thank you for the comments.  The sentences have rephrased as shown on line 103-107.

12. Lines 111-127: This paragraph should be carefully rewritten. For example, what is PVA?  Once “1% BSA in Dulbecco's phosphate-buffered saline-PVA” was abbreviated as BSA-DPBS-PVA, that should be used consistently. This reviewer guesses “Dulbecco's phosphate-buffered saline” should be consistently described as “DPBS”. What is happened in lines 118-119? Authors should more carefully write the manuscript for publication.

Response: The abbreviations have been defined and used thereafter (Line113-123 and Line131-137).

14. Line 138: “2.3” should be “2.4”.

Response: It has been corrected to “2.4”, and sorry for the mistake (Line 140).

15. Line 153: “Subs” should be “Substrate”.

Response: It has been revised as suggested (Line 154).

16. Section 2.5: Many sentences can be seen in section 2.2-2.4, thus they are repetitive. Section 2.2-2.5 can be reorganized and rewritten.

Response: Thank you for the comments.  These sections have been improved (Line158-176).

17. Figure 3: (a) and (b) should be A and B, respectively. What is “M” in Fig. 3B?

Response: Thank you for the help.  The mistakes have been corrected, and M stands for marker as added to the figure legend (We have switched Figure 3 and Figure 5).

18. Figure 5: (a) and (b) should be A and B, respectively.

Response: The mistakes have been corrected as shown in Figure 3 A and B. (We have change Figure 5 to Figure 3)

19. Discussion: The greater part of “Discussion”, e.g. 2nd and 3rd paragraph, cannot be tightly connected with results obtained in this report, which is confusing this reviewer very much. Discussion is alien to the results and conclusions.

      Response: Thanks for the comments. We have completely reorganized the order of Figure presentation (original Figures 3 and 5 are switched) and the context of Discussion as suggested (Line 291-349).

Reviewer 2

Open Review

English language and style

(x) Extensive editing of English language and style required 
( ) Moderate English changes required 
( ) English language and style are fine/minor spell check required 
( ) I don't feel qualified to judge about the English language and style 

Yes

Can be improved

Must be improved

Not applicable

Does the introduction provide sufficient background and include   all relevant references?

( )

(x)

( )

( )

Is the research design appropriate?

(x)

( )

( )

( )

Are the methods adequately described?

( )

(x)

( )

( )

Are the results clearly presented?

( )

( )

(x)

( )

Are the conclusions supported by the results?

( )

( )

(x)

( )

Comments and Suggestions for Authors

The study describes the expression of p-38 and phosphorylated p-38 in the porcine oocyte during in vitro maturation.  An increase in the level of p38 and p-p38 in the oocyte was observed throughout maturation from germinal vesicle (GV) to metaphase II (MII). Relative activity of p-38, as assessed through the ratio of activated (phosphorylated p-38) to total p-38, was lowest in GV stage oocytes and increased from GVBD.  Changes in the subcellular localization of p-p38 within the oocyte are also described.   The results are of some interest, and add to the understanding of the potential role of p-38 activation in oocyte maturation.  

General comments:

While these findings support a role for p-38 activation in regulating porcine oocyte maturation, the conclusion that they prove that it is essential (Line 39 and 319) seems a strong statement, given the nature of the study. 

It was not clear how novel the information on changes in nuclear chromatin configuration of the porcine oocyte during meiosis is.  Has anyone else reported the nuclear confirmation of GV stage porcine oocytes?  There appears to be minimal reference to other studies.

Response: Thank you for the comments.  Although oocyte cytoskeleton structures have been reported in previous studies both from our and some other groups, its patterns and classifications remained inconsistent. Therefore, these data mainly can serve as the confirmation of previous studies as well as the progression of oocyte maturation stages in this study. 

Methods: The authors have used methods they have experience with and have previously published.  The methods would benefit from a clear description of numbers of oocytes and replicates analysed.  These are reported in Figure legends, but should be incorporated into the methods and results.  The methods indicate that 200 oocytes were used for Western blotting, while the Figure legend reports 150 oocytes.

Response: Points are well taken, and sorry for the unnecessary mistake. We have corrected it to 150 oocytes and more info is added to our Materials and Methods as suggested (Line 170-171).

Results:  Line 211-213 appears to be discussing the immunoblot results, but is included in the section on subcellular localization. Should this sentence be in 3.2 rather than 3.3.

Response: We have changes section 3.2 (Subcellular localization of p-p38), section 3.3 (Expressions of p38 and p-p38 in porcine oocytes during IVM) and change Figure 3 to Figure 5 as shown on line 205-220.

Line 213-214: suggests that expression levels of total p-38 are shown in Figure 5b, but this figure reports p-p38, not p-38 expression?

Response: Thank you for the comments.  To avoid confusions, we have rephrased the sentence as shown on line 222-218 and line 213-219. We have change Figure 5b to Figure 3.

Line 215-216: suggests that Figure 5B reports the ratio of p-p38 / p-38, but this Figure reports p-38 expression, while Figure 3b reports the ratio.  This sentence also appears to relate to results from the immunoblot, and should be included in 3.2 rather than 3.3.

Response: Thank you for the comments.  To avoid confusions, we have rephrased the sentence as shown on line 205-220. (original Figures 3 and 5 are switched)

Discussion: The Discussion could be significantly improved.

Response: Points are well-taken and the Discussion has been dramatically reorganized. (Line 291-349)

The relevance of the final 2 sentences of the first paragraph is unclear (lines 269-272).  Oocytes from different size follicles were not studied.

Response: Thanks for pointing it out, and we have improved it in the first paragraph of the Discussion section (Line 291-305).

Line 277 appears to suggest that this study was the first to detect p-p38 in porcine oocytes by immunoblotting; however, the authors have previously used this method to detect expression in MII oocytes (Yen et al, 2014).

Response: “the first” has been deleted from the text (Line 291-305).

The third paragraph of the Discussion essentially replicates discussion from the authors previous paper (Yen et al 2014) on the expression and activation of MAPK in porcine oocytes under heat stress.  While some points have relevance to the current study, as culture could be another form of stress, it is unclear whether replication of the whole paragraph discussing heat shock proteins is warranted and appropriate.

Response: Thanks for pointing out the problem, but we have rephrased the sentences as possible to improve it, and also cite our previous published paper to avoid unnecessary arguments (Line 341-349).  

The discussion appears to end prematurely - Line 317 is an incomplete sentence.

Response: We have checked through and improved the text in the Discussion as suggested. (Line 307-321)

Other comments:

Line 20 and Line 34: It is not clear how subcellular localization of p-p38 can increase. Was the purpose of this sentence to comment on changes in the localization of p-p38, or to comment on the increase in p-p38 expression throughout oocyte maturation.

Response:Thank you for the comment.  we have revised “in the localization of p-p38” line 20 and the increase in p-p38 expression throughout oocyte maturation line 35-36.

Line 21: "Alteration of p38 activation" is unclear, please reword.

Response:  We have rewritten “(Alteration expression of p38 activation appears to participate in regulating oocyte IVM, along with the progressive reorganization of the cytoskeleton and redistribution of cytoplasmic p-P38.)” as shown in line 21-23.

Line 27: The level of activated p-38, please reword.

Response: We have changed it to “The levels of p-p38 or activated p-38 and p-38 expressions were….” as shown in Line 27-28.

Line 27: immunocytochemistry

Response: The mistakes have been corrected (Line 27).

Line 118: "containing anti- and" please provide the full name of this antibody.

Response: The mistakes have been corrected "containing anti-actin and….." as shown on line 121.

Line 129: Fixed oocytes, please correct spelling

Response: Sorry for the mistake and an “s” has been deleted as shown in line 130.

Line 137: expression in oocytes

Response: Thank you for the help.  The mistakes have been corrected " in" as shown on line 138.

Line 137: Please provide more information on the analysis used for determining the intensity of expression.  Was the whole oocyte assessed, were controls in place to ensure staining intensity was similar between replicates?

Response: Yes, all the intensity measurements used the whole oocyte. A control group was always in place for each batch of staining, plus a skillful controlled staining time to minimize variations among treatment groups. Although inadequate, we assumed that all the unavoidable errors should have been compromised by replicates.    

Line 202: what is meant by "after prolonged IVM culture", the period described appears to be the standard period for IVM of porcine oocytes.

Response: Sorry for the mistakes, but it has been changed to " during IVM culture " (Line 215-216).

Figure 3a, why is no marker visible for the p38 blot.

Response: Thank you for the help. We have replaced Figure 5A (We have also switch Figure 3 with Figure 5) with one showed the marker lane.

Line 297: "during the prolonged heat shock" please remove the word protein.

Response: Revised as suggested in discussion.

Round 2

Reviewer 2 Report

Most of the comments have been addressed.  

The experimental design indicates that subcellular localisation of both p38 and p-p38 was examined, however, this is not mentioned in the methods section.  Only the p-p38 antibody is mentioned in the methods section.

Sections 3.2 and 3.3 are still somewhat confusing.

Section 3.2 comments on total p38 expression in GV oocytes (line 193); however, Figure 3 shows p-p38 expression.  It is not clear whether Fig 3B is relative activity.  The text indicates that these results are a ratio of p-p-38/p38 (line 195).  However, the figure legend indicates that this this fluorescence intensity of p-p38 (line 241).  It is unclear whether subcellular localisation of p-38 was also studied, and used to calculate a ratio.

Section 3.3 suggests that Fig 5B shows p-p38 level, however, this Figure does appear to show the ratio. 

Minor comments:

Line 25: and plays

Line 45-46: which are involved in a variety of cellular responses to cellular cues ranging from regulation of the cell cycle, cell death....

Line 76: Microtubules are a dynamic

Line 98: gilts

Line 99: to the laboratory

Line 156: Oocytes were studied at the GV

Line 159: Expression of p38

Line 227: become thickened

Line 297: therefore, a linear increase in the expression of p-p38 levels in the ooplasm with progression of IVM was observed

Line 318: downstream of p38

Line 324: it is not clear what is meant by "oocyte maturation resemble MPF"

Line 335: regulation of porcine

Author Response

Open Review

English language and style

( ) Extensive editing of English language and style required 
(x) Moderate English changes required 
( ) English language and style are fine/minor spell check required 
( ) I don't feel qualified to judge about the English language and style 

Yes

Can be improved

Must be improved

Not applicable

Does the introduction provide sufficient background and   include all relevant references?

(x)

( )

( )

( )

Is the research design appropriate?

(x)

( )

( )

( )

Are the methods adequately described?

( )

(x)

( )

( )

Are the results clearly presented?

( )

(x)

( )

( )

Are the conclusions supported by the results?

(x)

( )

( )

( )

Comments and Suggestions for Authors

Most of the comments have been addressed.  

The experimental design indicates that subcellular localisation of both p38 and p-p38 was examined, however, this is not mentioned in the methods section.  Only the p-p38 antibody is mentioned in the methods section.

Response: We have improved it by adding ”p-38 analysis” after “Western blot” (L139); also, in the “2.4. Western Blotting” info for p-38 antibodies were added (L148).

Sections 3.2 and 3.3 are still somewhat confusing.

Section 3.2 comments on total p38 expression in GV oocytes (line 193); however, Figure 3 shows p-p38 expression.  It is not clear whether Fig 3B is relative activity.  The text indicates that these results are a ratio of p-p-38/p38 (line 195).  However, the figure legend indicates that this this fluorescence intensity of p-p38 (line 241).  It is unclear whether subcellular localisation of p-38 was also studied, and used to calculate a ratio.

Response: Sorry for the confusion; actually, we have switched previous Fig 3 with Fig 5 in the revised version. Therefore, the new Fig 3 is no longer the relative (ratios) p-p38/p-38 levels; instead, it is the p-p38 intensity or amount observed in the cytoplasm. Relative levels (fold changes) of p-p38/p-38 is now presented in Fig 5. 

Section 3.3 suggests that Fig 5B shows p-p38 level, however, this Figure does appear to show the ratio. 

Response: It has been changed as explained in the previous question for Section 3.2.

Minor comments:

Line 25: and plays

Response: Thanks; an “s” has been added to “play” (L25).

Line 45-46: which are involved in a variety of cellular responses to cellular cues ranging from regulation of the cell cycle, cell death....

Response: The mistakes have been corrected (LL45-46).

Line 76: Microtubules are a dynamic

Response: We have deleted and rewrite (LL98-83).

Line 98: gilts

Response: Revised as suggested (L104).

Line 99: to the laboratory

Response: Revised as suggested (L105).

Line 156: Oocytes were studied at the GV

Response: Revised as suggested (L163).

Line 159: Expression of p38

Response: The “shas been deleted as suggested (L166).

Line 227: become thickened

Response: Revised as suggested (L234).

Line 297: therefore, a linear increase in the expression of p-p38 levels in the ooplasm with progression of IVM was observed

Response: Revised as suggested (LL305-306).

Line 318: downstream of p38

Response: Revised as suggested (L327).

Line 324: it is not clear what is meant by "oocyte maturation resemble MPF"

Response: We have changed “resemble” to “similar to”  and corrected the mistakes in the sentence (LL334-335).

Line 335: regulation of porcine

Response: Revised as suggested (Line 345).